# A compact design of four-port high-gain MIMO antenna using hybrid coupler and dual-polarized radiators

Phuong Kim-Thi[ID][1]*, Thang Nguyen-Van[2], Dat Nguyen-Tien[3], Tung Bui-Thanh[ID][2]

**1** Faculty of Electrical and Electronics Engineering, Thuyloi University, Hanoi, Vietnam, **2** Faculty of Electronics and Telecommunications, VNU - University of Engineering and Technology, Hanoi, Vietnam, **3** Faculty of Electrical and Electronic Engineering, PHENIKAA School of Engineering, PHENIKAA University, Hanoi, Vietnam

* phuongkt@tlu.edu.vn

## Abstract

This paper introduces a design methodology for a multiple-input multiple-output (MIMO) antenna that achieves multi-port operation, high gain, as well as compact dimensions. The proposed design employs two dual-polarized radiators integrated with two hybrid couplers. This architecture enables a four-port MIMO array to be realized using only two radiating elements. Meanwhile, the hybrid couplers simultaneously excite both ports of the dual-polarized elements with equal magnitude and a 90° phase difference, facilitating high-gain radiation. To validate the proposed concept, an antenna prototype was fabricated and tested. Measurements confirm that the antenna, with an overall compact size of $0.96\lambda \times 0.77\lambda \times 0.04\lambda$, achieves a 3.1% operating bandwidth (4.72-4.87 GHz) with inter-port isolation better than 10 dB. Within this band, the antenna maintains a measured gain of around 8.0 dBi. Additionally, the antenna also performs good diversity performance in terms of Envelop Correlation Coefficient, Diversity Gain, Channel Capacity Loss, and Mean Effective Gain.

## Introduction

Multiple-input multiple-output (MIMO) antenna systems have garnered significant attention in modern wireless communication due to their ability to enhance data throughput and spectral efficiency without requiring additional frequency resources [1]. Additionally, as electronic devices are getting smaller and smaller while requiring high channel capacity and long-range communication, there is a strong demand for a MIMO array, which possesses multi-port operation, compact size, as well as high gain radiation.

The microstrip patch antenna has been demonstrated as an effective solution to achieve high gain with compact size structure. Various MIMO patch antennas have

**Data availability statement:** All relevant data are within the manuscript.

**Funding:** The author(s) received no specific funding for this work.

**Competing interests:** The authors have declared that no competing interests exist.

been reported in the open literature [2–6]. These conventional MIMO designs typically rely on single-polarized radiators, where each antenna port is connected to a separate radiating element. As the number of ports increases, this configuration leads to larger antenna footprints. For instance, four radiating elements are required for 4-port MIMO array. A possible solution to design multi-port MIMO with less radiators is to employ dual-polarized patches. This approach has been presented in [7–10], in which 4-port MIMO array can be realized with only two dual-polarized radiators. Nonetheless, the radiator utilized in [2–10] working in the fundamental $TM_{01}/10$ mode performs low-gain radiation, which is typically around 6 dBi. Using a high-order mode patch can increase the gain to about 9-10 dBi, but trade-off with extremely large antenna dimensions [11,12]. Similar gain performance can be achieved by using dielectric resonator structures [13,14].

Improving the gain of MIMO patch antennas is still a challenging task, especially when balancing compactness and the number of uncorrelated waves. The literature review indicates that gain enhancement in MIMO antennas has traditionally been achieved using frequency selective surfaces (FSS) [15–17], T-junction power dividers [18–20], or a combination of those approaches [21–23]. FSS layers placed above the radiators can enhance gain but at the cost of increased vertical profile. Meanwhile, T-junction-based techniques improve gain by exciting multiple elements per port, but this also demands more radiators and larger substrate areas accordingly. Additionally, it is worth noting that using these traditional techniques makes the antenna size extremely increase when multi-port MIMO operation is required.

To overcome these drawbacks, this paper proposes an approach to design MIMO antenna with high-gain, compact, and multi-port operation based on a combination of dual-polarized radiators and hybrid couplers. Dual-polarized radiators offer two orthogonal polarizations from a single radiator, effectively halving the number of required elements for a given number of ports. Furthermore, hybrid couplers enable simultaneous excitation of both radiators, resulting in high gain operation. The measurement demonstrates that a 4-port MIMO antenna with compact dimensions of $0.96\lambda \times 0.77\lambda \times 0.04\lambda$ and high gain of around 8.0 dBi can be realized using the proposed approach. This compact and efficient architecture is highly suitable for space-constrained IoT terminals.

## Dual-polarized radiator and hybrid coupler design
### Dual-polarized radiator

Fig 1 illustrates the geometrical configuration of the dual-polarized patch antenna. The antenna is fabricated on a single-layer Taconic RF-35 substrate characterized by a relative permittivity of 3.5 and a low loss tangent of 0.002. The radiating patch is orthogonally fed via two coaxial feed points, enabling independent excitation of two orthogonal linear polarizations vertical and horizontal. The optimal dimensions of the antenna are as follows: $W_s = 40$, $w = 15.7$, $d = 2.5$ and $h = 1.52$ (unit: mm).

Fig 2 depicts the simulated S-parameter (reflection coefficient $|S_{11}|$ and transmission coefficient $|S_{21}|$) and realized gain of the dual-polarized patch antenna. Obviously, the antenna shows good matching performance around the interested

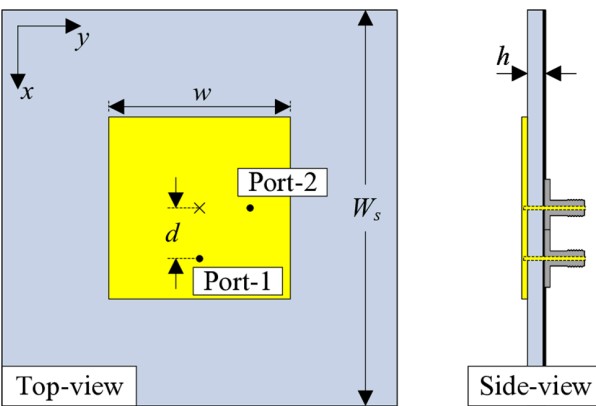

**Fig 1.** **Geometry of the dual-polarized patch.**

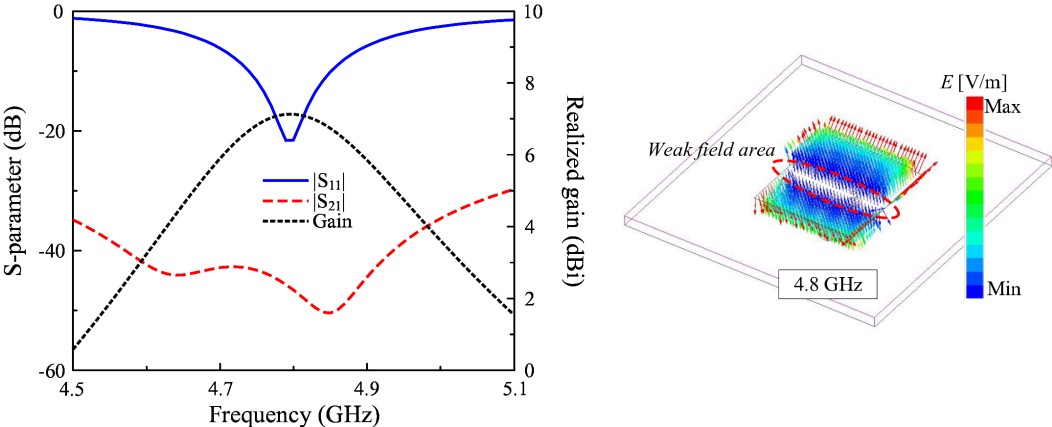

**Fig 2.** **Simulated performance of the dual-polarized patch.**

frequency band of 4.8 GHz, with reflection coefficient below -10 dB. Theoretically, the half-effective-wavelength square patch with dual feeding positions supports two dominant orthogonal modes, i.e., $TM_{10}$ and $TM_{01}$. When the antenna is excited through Port-1 (aligned along the x-axis), the field distribution establishes a null line characterized by zero electric potential along the orthogonal y-axis direction [20]. This field symmetry significantly suppresses energy coupling to the orthogonal Port-2, thereby achieving exceptionally high inter-port isolation. Here, the simulated isolation levels reach approximately 45 dB. Additionally, the simulated realized gain in the broadside direction at 4.8 GHz is about 7.1 dBi.

## Hybrid coupler

The geometry of a hybrid coupler operating at 4.8 GHz is presented in Fig 3. The utilized substrate is the FR-4 substrate with a dielectric constant of 4.4 and a loss tangent of 0.02. Basically, hybrid coupler is a four-port passive microwave device used to equally split an input signal into two outputs with a specific phase difference of typically 90°. Here, the power flows into Port-1 and then distributes to Port-2 and Port-3 with equal amplitude and 90° phase difference. Port-4 is an isolated port and ideally receives no power when Port-1 is fed. The line widths, $w_1$ and $w_2$, are chosen for their impedance of about 50 Ω and 35 Ω. The length l is about a quarter wavelength. The design parameters of the hybrid couple are as follows: $w_1 = 1.5$, $w_2 = 2.6$, $l = 7.5$, and $h_f = 0.8$ (unit: mm).

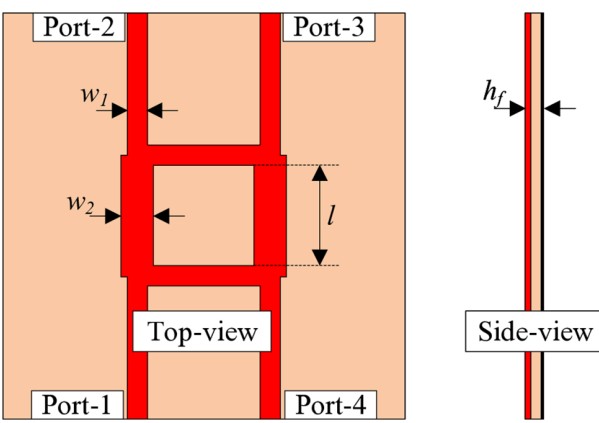

**Fig 3. Geometry of the hybrid coupler.**

The simulated performance in terms of magnitude and phase of the hybrid coupler is depicted in Fig 4. It is obvious that around 4.8 GHz, the port matching performance is very good, with a dip in return loss at approximately -50 dB. Meanwhile, the transmission coefficients to the output ports, $|S_{21}|$ and $|S_{31}|$, are nearly equal at approximately -3.5 dB, showing a balanced power split. The isolation between Port 1 and Port 4 is quite high at about 30 dB. With respect to the phase difference between Port 2 and Port 3, the simulated data indicates that the phase difference is around 90°.

## Four-port MIMO antenna

### Antenna design

Fig 5 shows the geometry of the proposed 4-port MIMO antenna, which is a combination of two dual-polarized radiators and two hybrid couplers presented in the previous section. Two couplers, designated as C-1 and C-2, have four input ports including P-1, -2, -3, and -4. These couplers are implemented on a low-cost FR-4 substrate, as simulations confirmed its performance was adequate for the feeding network, while the low-loss Taconic RF-35 substrate was reserved for the radiating elements to maximize efficiency. Two dual-polarized radiators named Ant-1 and Ant-2 are rotated at 45°

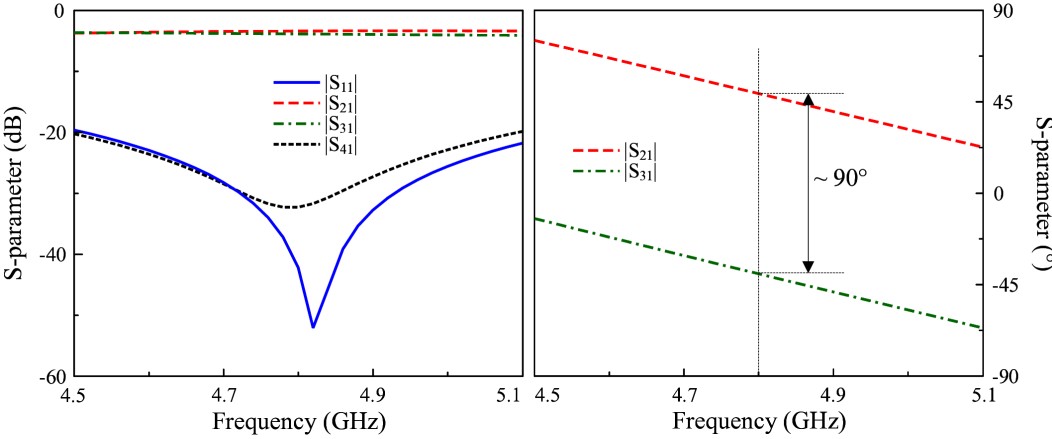

**Fig 4. Simulated S-parameter of the hybrid coupler.**

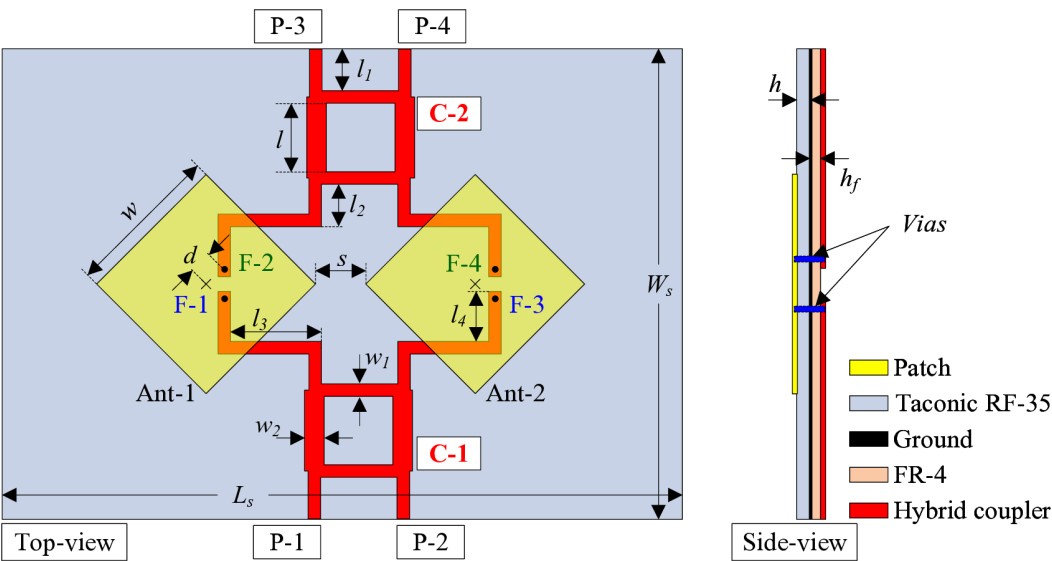

**Fig 5**. Geometry of the proposed 4-port MIMO antenna.

to ensure the symmetrical configuration for all port operations. The output ports of C-1 are F-1 and F-3, corresponding to dual feeding positions on Ant-1 and Ant-2. Noted that the polarizations of Ant-1 and Ant-2 with feeding positions at F-1 and F-3 are identical. The optimized dimensions of the proposed 4-port MIMO antenna are $L_s = 70$, $W_s = 48$, $h = 1.52$, $h_f = 0.8$, $w = 15.6$, $d = 2.4$, $w_1 = 1.5$, $w_2 = 2.3$, $l = 6.9$, $l_1 = 4$, $l_2 = 4.3$, $l_3 = 9.2$, and $s = 6$ (unit: mm).

## Antenna performance

The simulated S-parameter and peak realized gain of the proposed 4-port MIMO antenna are depicted in Fig 6. Noted that the S-parameter for all ports is identical due to the symmetrical geometry, only the result for Port-1 operation is chosen to be shown. As observed, the antenna exhibits good performance around 4.8 GHz, in which the reflection coefficient and transmission coefficients are all below -10 dB. Meanwhile, the peak realized gain values are about 8.5 dBi within the operating frequency range from 4.74 to 4.85 GHz. In comparison with the dual-polarized antenna discussed in the previous section, the gain improvement is about 1.4 dBi. Additionally, the simulated radiation patterns at 4.8 GHz for different operating ports, as illustrated in Fig 7, show that the main beam is tilted off the broadside direction due to the different excitation phases (90°) of the radiating elements. The main beam direction is similar for P-1, P-3 and P-2, P-4.

## Antenna operation characteristics

As described above, the design exploits the coupler as a power divider to excite both elements with similar magnitude. The operation can be depicted in Fig 8. Here, both antenna elements radiate with high aperture efficiency. Therefore, high gain radiation can be attained. This is different from typical 2-element MIMO systems where each element radiates separately. Meanwhile, the isolation is now controlled by the isolation of the coupler and the reflection coefficient of each antenna element. The diagrams demonstrate that perfect isolation of the system is achieved when the reflection coefficient is zero ($S_{11}(Ant-1) = 0$ and $S_{11}(Ant-2) = 0$) and the isolation of the coupler is perfect ($S_{21}(C) = 0$). These two conditions are relatively easy to achieve by optimizing the coupler and the antenna separately. For the proposed 4-port MIMO design, the feeding positions of F-1 and F-3 will strongly affect the isolation between P-1 and P-2. Meanwhile, due to the high isolation between F-1 and F-2, F-3 and F-4 (demonstrated in Sect 2), the isolation between P-1 and P-3, P-4 will be

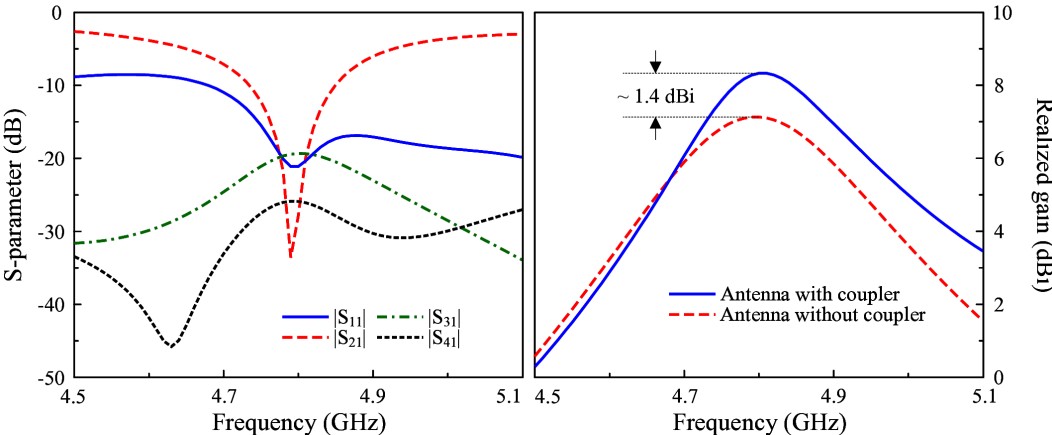

**Fig 6**. Simulated S-parameter and realized gain of the proposed 4-port MIMO antenna.

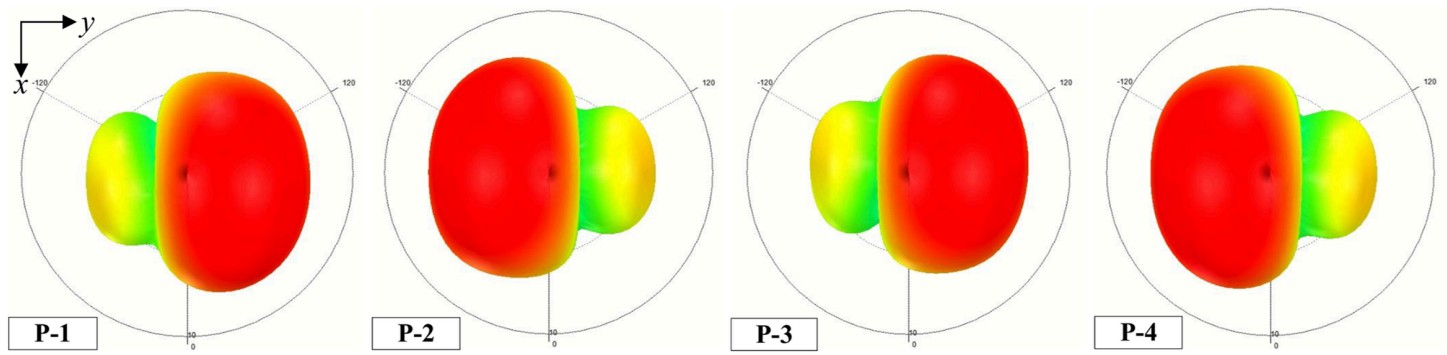

**Fig 7**. Simulated radiation patterns at 4.8 GHz of the proposed 4-port MIMO antenna.

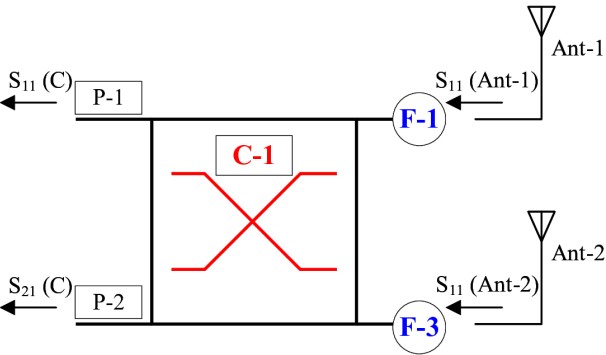

**Fig 8**. Diagram of the antenna working with hybrid coupler.

high. Further demonstration can be observed in Fig 9, which shows the S-parameter of the proposed antenna with P-1 excitation with different feeding positions, d. Obviously at 4.8 GHz, changing d strongly affects the matching performance, and when the mismatch is worse, the isolation is significantly degraded.

The simulated current distribution on the power dividers and radiating elements are illustrated in Fig 10. With P-1 excitation, the power is distributed to the two output ports to excite both radiating elements simultaneously, demonstrating the high gain performance. Meanwhile, the power distribution on the other input ports of the couplers (P-2, -3, and -4) is quite weak. This demonstrates the high isolation between the input ports.

Element spacing plays a critical role in this antenna design, as it directly influences both the radiation performance and the compactness of the structure. The impedance matching and isolation can be fine-tuned by adjusting the hybrid dimensions and feeding position. Fig 11 shows the simulated gain radiation patterns at 4.8 GHz for different element spacing, $s$. As observed, the gain of the main lobe is slightly changed with the variation of $s$. However, larger element spacing leads to the emergence of higher grating lobes. When the distance is much larger than the ideal distance of half wavelength, it significantly degrades radiation pattern integrity.

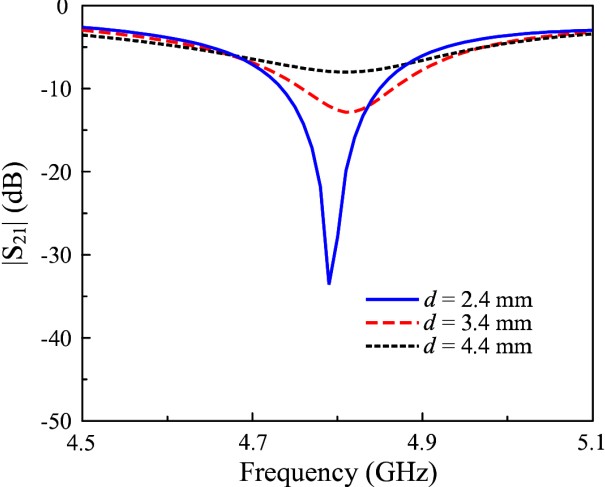

**Fig 9. Simulated $|S_{21}|$ for different feeding positions, d.**

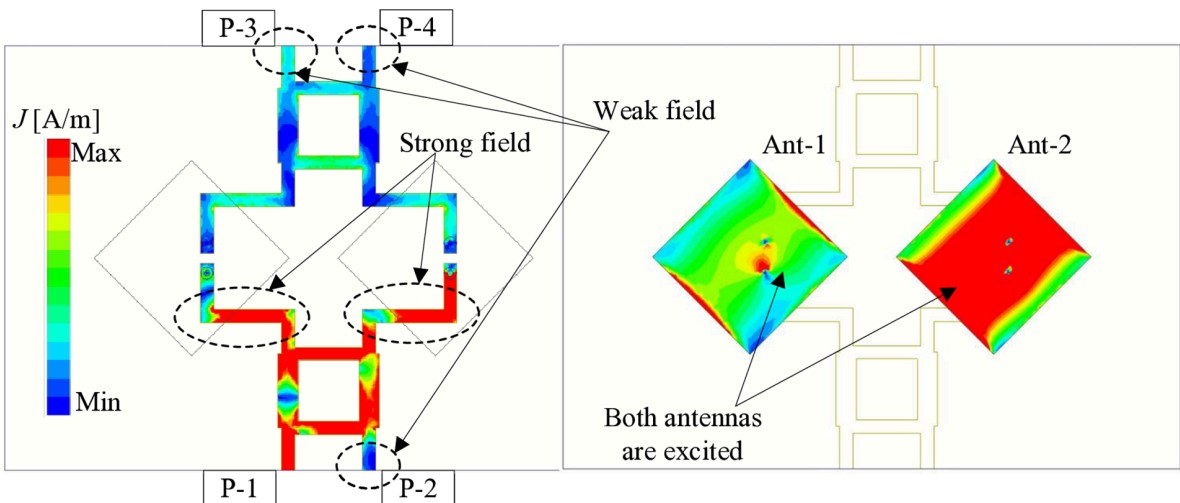

**Fig 10. Simulated current distribution at 4.8 GHz.**

## MIMO diversity performance

The MIMO diversity performance evaluated by Envelop Correlation Coefficient (ECC), Diversity Gain (DG), Channel Capacity Loss (CCL), and Mean Effective Gain (MEG) is investigated [24]. The ECC measures the correlation between radiation patterns of two MIMO antenna ports and the ECC smaller than 0.01 demonstrates the high-performance system. The DG represents the gain in signal-to-noise ratio (SNR) due to diversity compared to a single antenna system and the ideal value of DG is close to 10 dB. Meanwhile, the CCL shows maximum data rate that can be reliably transmitted using the MIMO system. This is computed based on S-parameter and the acceptable value of ECC is less than 0.4 *bps/Hz*. Finally, the MEG parameter shows a ratio of the average received power of each port to the total available power under a specific angular power distribution and MEG ratio of less than 3 dB is preferred. The calculated equations for these parameters can be found in [25]. The calculated ECC, DG, CCL, as well as MEG are presented in Figs 12 and 13. The data indicates that all calculated results are confined to acceptable values, which demonstrate good diversity performance of the proposed 4-port MIMO antenna.

## Measurement results

To experimentally validate the proposed design, a 4-port MIMO antenna prototype, as depicted in Fig 14, was fabricated and subjected to comprehensive measurements. Overall, the measured results exhibit strong agreement with full-wave simulation data, confirming the accuracy of the design approach. Minor discrepancies observed between simulated and measured results are attributed to fabrication tolerances, variations in the substrate's dielectric constant, connector losses, and potential imperfections in the measurement setup.

Fig 15 presents the simulated and measured S-parameters for P-1 excitation of the proposed 4-port MIMO antenna. The results confirm a common operating bandwidth (BW) ranging from 4.72 to 4.87 GHz, within which all ports exhibit return losses below -10 dB and inter-port isolation consistently exceeding 10 dB. There is a small difference with the simulated results, in which the operating BW ranges from 4.74 to 4.85 GHz. Due to the symmetrical structure of the antenna, the measured S-parameter results of other ports are quite similar to those of P-1. Accordingly, the far-field radiation patterns for P-1 excitation are representative and thus selected for analysis.

Fig 16 plots the gain radiation patterns in the principal E- and H- planes at 4.8 GHz. The main lobe of the proposed antenna is tilted approximately 25° off the broadside, which is due to the 90° phase difference introduced by the hybrid

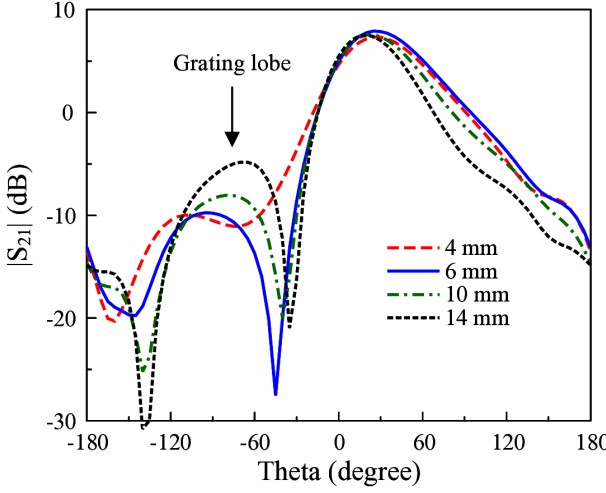

**Fig 11**. Simulated gain radiation patterns at 4.8 GHz for different element spacing, s.

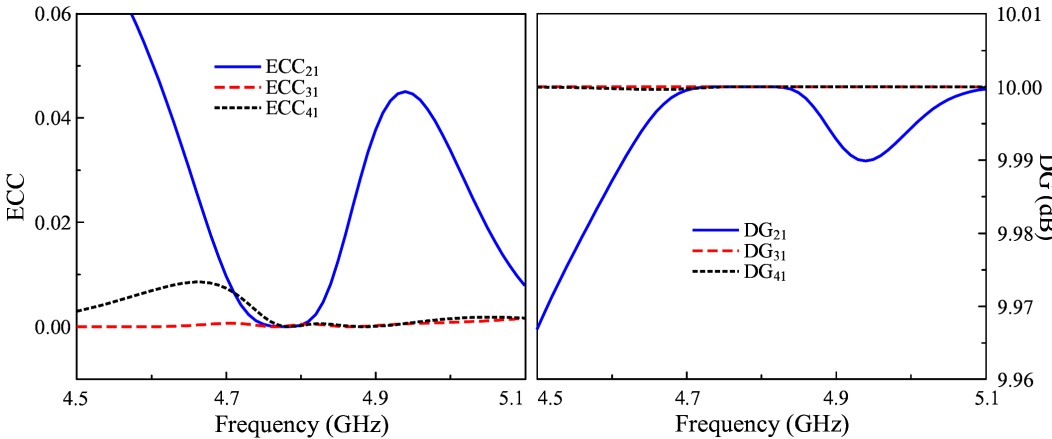

**Fig 12. Calculated ECC and DG of the proposed 4-port MIMO antenna.**

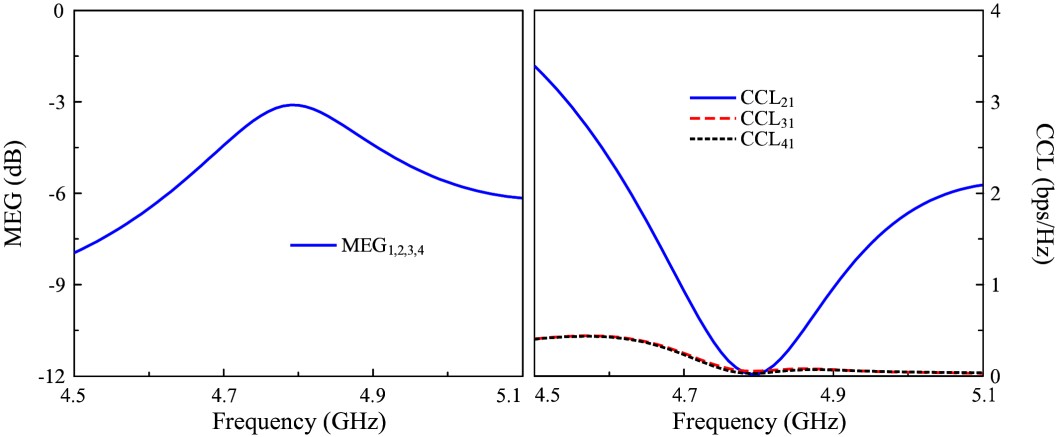

**Fig 13. Calculated CCL and MEG of the proposed 4-port MIMO antenna.**

coupler. The measured maximum gain of about 8.0 dBi can be obtained in the direction of $\theta = 25°$. Additionally, the cross-polarization radiation on this direction is about 23 dB smaller than the co-polarization radiation. This enhances communication reliability in multi-path environments. The measured cross-polarization is observed to be higher than the simulated levels. This discrepancy is likely caused by the influence of the coaxial feeding cables, whose placement can introduce field asymmetries, as well as minor unavoidable misalignments during the manual fabrication and assembly process.

A performance comparison among various MIMO design approaches is presented in Table 1. Here, different gain-enhancement techniques are involved in the table. Obviously, the method of using single or dual-polarized patches as MIMO elements has drawback of low gain. Among the gain enhancement techniques, while FSS-based approaches are effective in enhancing gain, they significantly increase the antenna's vertical profile. On the other hand, T-divider-based architecture necessitates a greater number of radiating elements to achieve high gain performance, thereby resulting in larger and more cumbersome antenna arrays. When taking overall size, number of MIMO ports, as well as gain radiation into account, the proposed method obviously requires the smallest number of radiating elements, while achieving the largest number of MIMO ports and comparable gain.

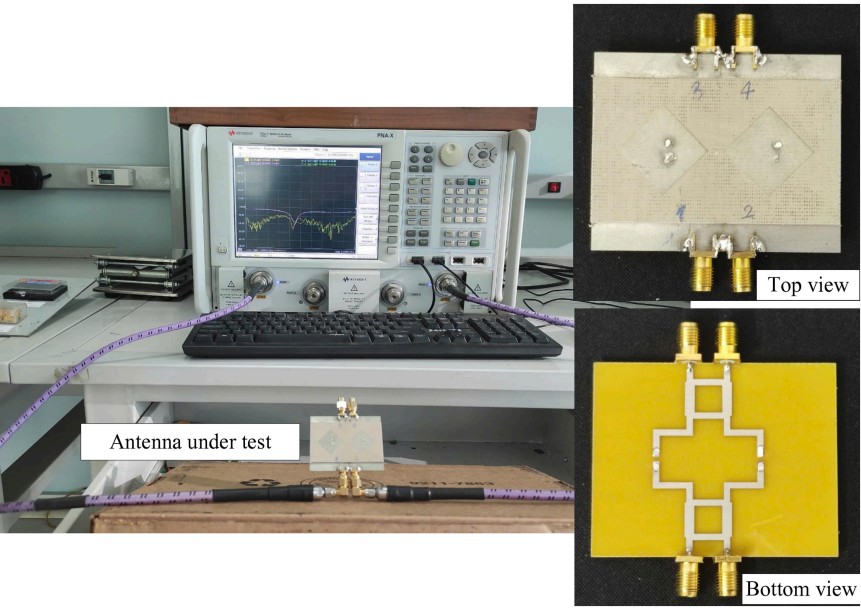

**Fig 14**. **Fabricated antenna prototype.**

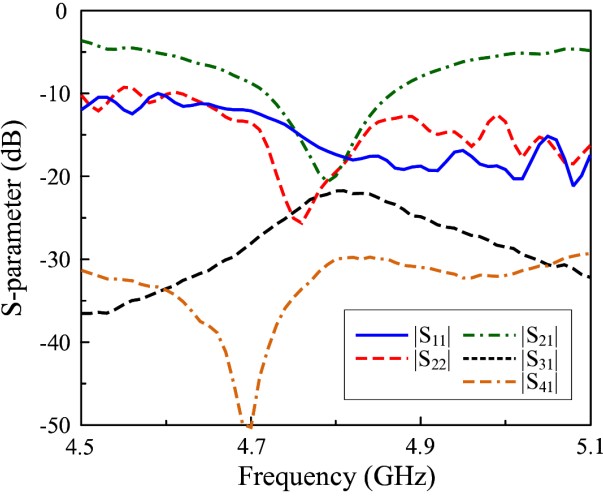

**Fig 15**. **Measured P-1 S-parameter of the proposed antenna.**

## Conclusion

This work presents a design methodology for a MIMO antenna system that simultaneously achieves multi-port operation, high gain performance, and compact physical dimensions. The proposed configuration integrates two dual-polarized patch antennas with two 90-degree hybrid couplers, thereby realizing a four-port high-gain MIMO array with only two radiating elements. To experimentally validate the concept, a prototype with compact dimensions of $0.96\lambda \times 0.77\lambda \times 0.04\lambda$ was fabricated and measured. The measured results achieve an operating BW of 3.1% (4.72-4.87 GHz), while maintaining inter-port isolation better than 10 dB and maximum gain of approximately 8.0 dBi. The data validates the suitability of the

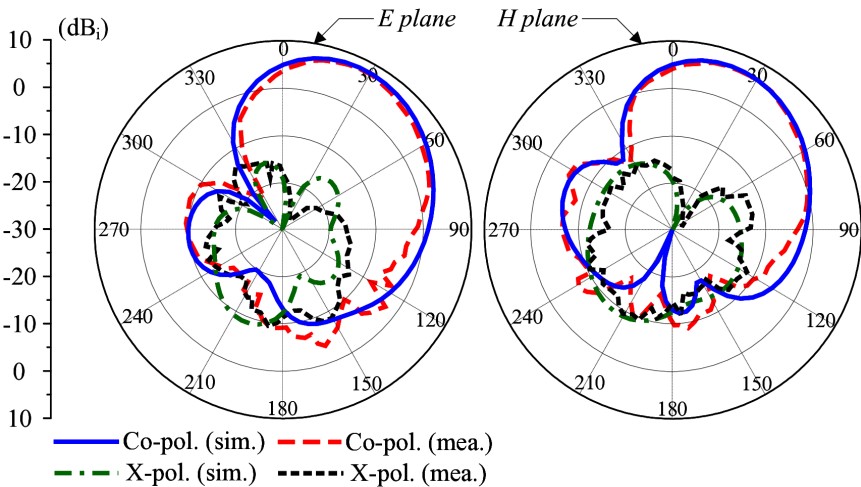

**Fig 16**. Simulated and measured radiation patterns at 4.8 GHz with P-1 excitation.

**Table 1**. Performance comparison among various MIMO antenna design approaches.

| Ref. | Overall size ($\lambda$) | MIMO configuration | No. of elements | No. of ports | BW (%) | Gain (dBi) |
|---|---|---|---|---|---|---|
| [4] | 0.69 × 0.52 × 0.02 | Single-pol. Patch | 2 | 2 | 2.5 | 3.8 |
| [5] | 0.88 × 0.58 × 0.03 | Single-pol. Patch | 2 | 2 | | 6 |
| [8] | 1.09 × 1.09 × 0.11 | Dual-pol. Patch | 4 | 8 | 12.8 | 6 |
| [10] | 2.94 × 2.94 × 0.09 | Dual-pol. Patch | 4 | 8 | 4.1 | 5.3 |
| [15] | 2.07 × 2.07 × 0.10 | Single-pol. Patch + FSS | 2 | 2 | 3.5 | 9.1 |
| [17] | 1.84 × 1.84 × 0.25 | Single-pol. Patch + FSS | 2 | 2 | 2.9 | 8.8 |
| [18] | 3.27 × 2.80 × 0.05 | Single-pol. Patch + T-divider | 8 | 4 | 3.5 | 11.5 |
| [20] | 2.24 × 2.24 × 0.07 | Single-pol. Patch + T-divider | 8 | 4 | 15.9 | 10.3 |
| [21] | 3.56 × 2.44 × 0.50 | Single-pol. Patch + T-divider + FSS | 8 | 4 | 7.6 | 10.3 |
| [22] | 2.14 × 1.16 × 0.16 | Single-pol. Patch + T-divider + FSS | 8 | 2 | 12.5 | 14.1 |
| Prop. | 0.96 × 0.77 × 0.04 | Dual-pol. Patch + Coupler | 2 | 4 | 3.1 | 8 |

proposed antenna for high-efficiency, space-constrained MIMO applications. Besides, the proposed work can be further developed for massive MIMO systems.

## Author contributions

**Investigation:** Phuong Kim-Thi.

**Methodology:** Thang Nguyen-Van.

**Supervision:** Tung Bui-Thanh.

**Validation:** Dat Nguyen-Tien.

**Writing – original draft:** Phuong Kim-Thi.

**Writing – review & editing:** Tung Bui-Thanh.

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
