## [Decision Letter · Decision Letter 0]

17 Sep 2025

PONE-D-25-38799A compact design of four-port high-gain MIMO antenna using hybrid coupler and dual-polarized radiatorsPLOS ONE

Dear Dr. Kim-Thi,

Thank you for submitting your manuscript to PLOS ONE. After careful consideration, we feel that it has merit but does not fully meet PLOS ONE’s publication criteria as it currently stands. Therefore, we invite you to submit a revised version of the manuscript that addresses the points raised during the review process.

We look forward to receiving your revised manuscript.

Kind regards,

Sachin Kumar, Ph.D.

Academic Editor

PLOS ONE

Journal Requirements:

**Additional Editor Comments:**

The reviewers have provided positive feedback on your submitted manuscript and acknowledged its potential contribution. However, they have also raised certain concerns that need to be addressed in order to further improve the quality of the manuscript. We request you to carefully address all reviewer comments, revise the manuscript accordingly, and resubmit it for further evaluation.

Reviewers' comments:

Reviewer's Responses to Questions

**Comments to the Author**

1. Is the manuscript technically sound, and do the data support the conclusions?

Reviewer #1: Yes

Reviewer #2: Yes

2. Has the statistical analysis been performed appropriately and rigorously?

Reviewer #1: Yes

Reviewer #2: N/A

3. Have the authors made all data underlying the findings in their manuscript fully available?

Reviewer #1: Yes

Reviewer #2: Yes

4. Is the manuscript presented in an intelligible fashion and written in standard English?

Reviewer #1: Yes

Reviewer #2: Yes

5. Review Comments to the Author

Reviewer #1: This manuscript presents a well-defined and valuable contribution to the field of MIMO antenna design. The authors propose a novel method for achieving a compact, high-gain, four-port MIMO antenna by integrating two dual-polarized radiators with two hybrid couplers. This approach cleverly uses the couplers to excite both radiating elements simultaneously from a single input port, thus achieving high gain while realizing a four-port system with only two radiating elements. The paper is well-structured, the methodology is sound, and the results are compelling. The experimental validation through a fabricated prototype confirms the viability of the proposed concept.

- The measured S-parameter results in Figure 14 show a slight frequency shift downwards compared to the simulated data. This is a very common effect and the authors' explanation of fabrication tolerance is appropriate. A brief sentence explicitly acknowledging this shift in the main text would show a thorough analysis of the results.

- The x-axis label in Figure 10, which displays gain radiation patterns, is incorrectly labelled as "Frequency (GHz)". This axis should represent the spatial angle in degrees (e.g., "Angle (degrees)" or " θ (degrees)"). Please correct this label.

- There is a notable inconsistency in the reported operating bandwidth. The Abstract states a bandwidth of 3.1% (4.72-4.87 GHz). However, the Conclusion reports a measured operating bandwidth of 2.1% (4.72-4.87 GHz). This is a significant discrepancy that must be corrected for consistency throughout the manuscript. Please verify update the text accordingly in all relevant sections.

- What was the primary reason for using FR-4 for the couplers instead of fabricating the entire device on the higher-performance Taconic substrate? Was this choice driven by cost, specific performance requirements for the coupler, or other manufacturing considerations?

- Regarding Figure 15, the measured cross-polarization is significantly higher than the simulated result. Beyond general tolerances, could you speculate on more specific causes?

Reviewer #2: The authors have proposed A compact design of four-port high-gain MIMO antenna using hybrid coupler and dual-polarized radiators. The following are the suggestions/observations

• The abstract should have all the antenna parameters.

• The gain values are expected to be mentioned in dBi across the manuscript.

• The introduction lacks depth significantly. You may refer the following articles along with other articles to enhance the depth.

o Enhanced isolation in aperture fed dielectric resonator MIMO antennas for 5G Sub 6 GHz applications. Scientific Reports, 15(1), p.10653.

o Substrate integrated waveguide fed dual band quad-elements rectangular dielectric resonator MIMO antenna for millimeter wave 5G wireless communication systems. AEU-international Journal of Electronics and Communications, 137, p.153821.

• The E field presentation needs to be better.

• Please check the S-parameter graph. The return loss at 4.7 GHz is going below the graph points

• If feasible, a normalized radiation pattern may be kept. Please improve the explanation of the radiation patterns.

• Please mention how 3D polar plots help in understanding gain in reference to 2D plots.

• The MIMO diversity parameters explanation needs to be strengthened.

• The result explanation is significantly lacking in depth.

• The conclusion should have a future scope included.

• Please remove obsolete references.

6. PLOS authors have the option to publish the peer review history of their article (what does this mean?). If published, this will include your full peer review and any attached files.

Reviewer #1: No

Reviewer #2: No

---

## [Author Response · Author response to Decision Letter 1]

3 Nov 2025

Submission ID: PONE-D-25-38799

Original Article Title: “A compact design of four-port high-gain MIMO antenna using hybrid coupler and dual-polarized radiators”

To: Reviewer

Re: Response to reviewer

Dear Reviewer,

We appreciate you for your precious time in reviewing our paper and providing valuable comments. It was your valuable and insightful comments that led to possible improvements in the current version. The authors have carefully considered the comments and tried our best to address every one of them.

We are uploading our point-by-point response to the comments, an updated manuscript with red highlighting indicating changes, and a manuscript without track changes.

Best regards,

Reviewer 1: This manuscript presents a well-defined and valuable contribution to the field of MIMO antenna design. The authors propose a novel method for achieving a compact, high-gain, four-port MIMO antenna by integrating two dual-polarized radiators with two hybrid couplers. This approach cleverly uses the couplers to excite both radiating elements simultaneously from a single input port, thus achieving high gain while realizing a four-port system with only two radiating elements. The paper is well-structured, the methodology is sound, and the results are compelling. The experimental validation through a fabricated prototype confirms the viability of the proposed concept.

Concern #1: The measured S-parameter results in Figure 14 show a slight frequency shift downwards compared to the simulated data. This is a very common effect and the authors' explanation of fabrication tolerance is appropriate. A brief sentence explicitly acknowledging this shift in the main text would show a thorough analysis of the results.

Author response: Thank you for this excellent suggestion. We agree that explicitly mentioning the frequency shift improves the analysis. As suggested, we have added a sentence to the second paragraph of “Measurement results” section to acknowledge this observation.

Author action: The last sentence of the first paragraph in Measurement results section is rewritten with an additional phrase to describe the cause of simulation and measurement discrepancies.

Concern #2: The x-axis label in Figure 10, which displays gain radiation patterns, is incorrectly labelled as "Frequency (GHz)". This axis should represent the spatial angle in degrees (e.g., "Angle (degrees)" or " θ (degrees)"). Please correct this label.

Author response: We sincerely thank the reviewer for identifying this error. The x-axis label in Figure 10 was indeed incorrect and was a typographical error.

Author action: We have corrected the x-axis label in the revised Figure 10. The label has been changed from "Frequency (GHz)" to "Theta (degrees)" to accurately represent the radiation pattern plot.

Concern #3: There is a notable inconsistency in the reported operating bandwidth. The Abstract states a bandwidth of 3.1% (4.72-4.87 GHz). However, the Conclusion reports a measured operating bandwidth of 2.1% (4.72-4.87 GHz). This is a significant discrepancy that must be corrected for consistency throughout the manuscript. Please verify update the text accordingly in all relevant sections.

Author response: We sincerely thank the reviewer for identifying this critical inconsistency. The value of 2.1% in the Conclusion was indeed a typo error.

Author action: The bandwidth value in the Conclusion has been modified.

Concern #4: What was the primary reason for using FR-4 for the couplers instead of fabricating the entire device on the higher-performance Taconic substrate? Was this choice driven by cost, specific performance requirements for the coupler, or other manufacturing considerations?

Author response: Thank you for this insightful question regarding our design choice. The decision to use FR-4 for the hybrid couplers while using Taconic RF-35 for the radiators was driven by a strategic balance between cost and performance.

Cost-effectiveness: The primary reason was to reduce the overall fabrication cost. FR-4 is significantly more affordable than high-frequency laminates like Taconic RF-35. Using it for the feeding network layer makes the design more commercially viable.

Performance trade-off: We performed simulations which confirmed that for the compact size of the hybrid couplers operating at 4.8 GHz, the additional insertion loss from the FR-4 substrate was minimal and had an acceptable impact on the antenna’s overall performance. Conversely, the radiating elements benefit much more significantly from the low-loss properties of the Taconic substrate, as this directly maximizes their radiation efficiency and gain.

We have added a brief clarification in the manuscript to make this design rationale clear to the reader.

Author action: The following sentence is added to the first paragraph of subsection Antenna design:

“These couplers are implemented on a low-cost FR-4 substrate, as simulations confirmed its performance was adequate for the feeding network, while the low-loss Taconic RF-35 substrate was reserved for the radiating elements to maximize efficiency.”

Concern #5: Regarding Figure 15, the measured cross-polarization is significantly higher than the simulated result. Beyond general tolerances, could you speculate on more specific causes?

Author response: We would like to thank Reviewer for this insightful comment. We acknowledge the discrepancy between the simulated and measured cross-polarization levels. While fabrication tolerances contribute, we agree that a more specific explanation is warranted. We believe the higher measured cross-polarization likely stems from a combination of the following factors:

Effect of feeding cables: This is the most probable cause. In the measurement setup, the coaxial cables connected to the four ports can disrupt the field symmetry. Common-mode currents can be excited on the outer shield of these cables, which then act as unintentional radiators, degrading the polarization purity. This effect is challenging to model perfectly in simulation.

Assembly and fabrication asymmetries: Minor asymmetries during fabrication and manual assembly, such as a slight rotational misalignment of the patch radiators or inconsistencies in the soldering of the coaxial feed probes, can disrupt the ideal orthogonal current modes, leading to increased coupling between them and thus higher cross-polarization.

Measurement environment: Although conducted in an anechoic chamber, minor reflections or scattering from the antenna positioning equipment could also contribute to the measured cross-polarization levels.

We have added a brief discussion of these points in the manuscript to provide a more thorough analysis of the results.

Author action: The following sentences are added to the third paragraph of the section Measurement results: The measured cross-polarization is observed to be higher than the simulated levels. This discrepancy is likely caused by the influence of the coaxial feeding cables, whose placement can introduce field asymmetries, as well as minor unavoidable misalignments during the manual fabrication and assembly process.

Reviewer 2: The authors have proposed A compact design of four-port high-gain MIMO antenna using hybrid coupler and dual-polarized radiators. The following are the suggestions/observations.

Concern #1: The abstract should have all the antenna parameters.

Author response: Agreed.

Author action: More antenna parameters have been included in Abstract of the revised manuscript.

Concern #2: The gain values are expected to be mentioned in dBi across the manuscript.

Author response: Agreed.

Author action: The unit of gain has been modified in the revised manuscript.

Concern #3: The introduction lacks depth significantly. You may refer the following articles along with other articles to enhance the depth.

a. Enhanced isolation in aperture fed dielectric resonator MIMO antennas for 5G Sub 6 GHz applications. Scientific Reports, 15(1), p.10653.

b. Substrate integrated waveguide fed dual band quad-elements rectangular dielectric resonator MIMO antenna for millimeter wave 5G wireless communication systems. AEU-international Journal of Electronics and Communications, 137, p.153821.

Author response: The authors would like to thank the Reviewer for the constructive comment.

Author action: The suggested references have been included in the revised manuscript as references [13, 14]. Additionally, further discussion is also added to Paragraph 2, Section “Introduction”.

Concern #4: The E field presentation needs to be better.

Author response: Agreed.

Author action: The figure quality is enhanced in the revised manuscript.

Concern #5: Please check the S-parameter graph. The return loss at 4.7 GHz is going below the graph points.

Author response: The operating frequency of the proposed antenna ranges from 4.72 to 4.87 GHz. Thus, the |S41| at 4.7 GHz is out of this range, and the author believes that this value might not be important.

Concern #6: If feasible, a normalized radiation pattern may be kept. Please improve the explanation of the radiation patterns.

Author response: The authors concur with the Reviewer that normalized radiation patterns provide a clearer comparison between co- and cross-polarization components. However, since normalized patterns do not include actual gain values, the authors consider the use of real radiation patterns to be more appropriate in this case.

Concern #7: Please mention how 3D polar plots help in understanding gain in reference to 2D plots.

Author response: The 2D plots (as shown in Fig. 14) are more useful than 3D plots with respect to the co- and cross-polarization radiation. Fig. 6b shows 3D plots for better observation of different beam directions for different MIMO ports.

Concern #8: The MIMO diversity parameters explanation needs to be strengthened.

Author response: The authors would like to thank the Reviewer for the constructive comment. As the MIMO diversity performance in terms of ECC, DG, CCL, and MEG has been thoroughly investigated in [R1], the authors just briefly discussed the threshold values and meaning of each parameter in the manuscript.

[R1] https://doi.org/10.1016/j.aeue.2020.153361

Author action: Ref [R1] is added to the revised manuscript as ref [22].

Concern #9: The result explanation is significantly lacking in depth.

Author response: The authors would like to thank the Reviewer for the constructive comment.

Author action: Further experiment result discussion is added to the revised manuscript.

Concern #10: The conclusion should have a future scope included.

Author response: Agreed.

Author action: A future scope is added to Conclusion of the revised manuscript.

Concern #11: Please remove obsolete references.

Author response: The authors agree with the Reviewer that citing more recent works is more appropriate. Accordingly, all references in this paper are noteworthy, and the authors have selected studies published from 2018 onward. Only reference [1] is considered outdated.

Author action: Ref [1] is replaced in the revised manuscript.

---

## [Decision Letter · Decision Letter 1]

7 Nov 2025

A compact design of four-port high-gain MIMO antenna using hybrid coupler and dual-polarized radiators

PONE-D-25-38799R1

Dear Dr. Kim-Thi,

We’re pleased to inform you that your manuscript has been judged scientifically suitable for publication and will be formally accepted for publication once it meets all outstanding technical requirements.

Kind regards,

Sachin Kumar, Ph.D.

Academic Editor

PLOS ONE

Additional Editor Comments (optional):

Reviewers' comments:

Reviewer's Responses to Questions

**Comments to the Author**

1. If the authors have adequately addressed your comments raised in a previous round of review and you feel that this manuscript is now acceptable for publication, you may indicate that here to bypass the “Comments to the Author” section, enter your conflict of interest statement in the “Confidential to Editor” section, and submit your "Accept" recommendation.

Reviewer #1: All comments have been addressed

Reviewer #2: All comments have been addressed

2. Is the manuscript technically sound, and do the data support the conclusions?

Reviewer #1: Yes

Reviewer #2: (No Response)

3. Has the statistical analysis been performed appropriately and rigorously?

Reviewer #1: Yes

Reviewer #2: N/A

4. Have the authors made all data underlying the findings in their manuscript fully available?

Reviewer #1: Yes

Reviewer #2: Yes

5. Is the manuscript presented in an intelligible fashion and written in standard English?

Reviewer #1: Yes

Reviewer #2: Yes

6. Review Comments to the Author

Reviewer #1: (No Response)

Reviewer #2: The review comments have been adequately addressed. No further revisions/modifications are required.

7. PLOS authors have the option to publish the peer review history of their article (what does this mean?). If published, this will include your full peer review and any attached files.

Reviewer #1: No

Reviewer #2: No

---

## [Editor Report · Acceptance letter]

PONE-D-25-38799R1

PLOS ONE

Dear Dr. Kim-Thi,

I'm pleased to inform you that your manuscript has been deemed suitable for publication in PLOS ONE. Congratulations! Your manuscript is now being handed over to our production team.

Kind regards,

on behalf of

Dr. Sachin Kumar

Academic Editor

PLOS ONE